# Of Keeping and Tipping the Balance: Host Regulation and Viral Modulation of IRF3-Dependent *IFNB1* Expression

**DOI:** 10.3390/v12070733

**Published:** 2020-07-07

**Authors:** Hella Schwanke, Markus Stempel, Melanie M. Brinkmann

**Affiliations:** 1Institute of Genetics, Technische Universität Braunschweig, 38106 Braunschweig, Germany; hella.schwanke@helmholtz-hzi.de (H.S.); markus.stempel@helmholtz-hzi.de (M.S.); 2Viral Immune Modulation Research Group, Helmholtz Centre for Infection Research, 38124 Braunschweig, Germany

**Keywords:** interferon regulatory factor, IRF3, innate immunity, antiviral response, immune modulation, viral evasion, antagonist, interferon beta, enhanceosome, NF-κB

## Abstract

The type I interferon (IFN) response is a principal component of our immune system that allows to counter a viral attack immediately upon viral entry into host cells. Upon engagement of aberrantly localised nucleic acids, germline-encoded pattern recognition receptors convey their find via a signalling cascade to prompt kinase-mediated activation of a specific set of five transcription factors. Within the nucleus, the coordinated interaction of these dimeric transcription factors with coactivators and the basal RNA transcription machinery is required to access the gene encoding the type I IFN IFNβ (*IFNB1*). Virus-induced release of IFNβ then induces the antiviral state of the system and mediates further mechanisms for defence. Due to its key role during the induction of the initial IFN response, the activity of the transcription factor interferon regulatory factor 3 (IRF3) is tightly regulated by the host and fiercely targeted by viral proteins at all conceivable levels. In this review, we will revisit the steps enabling the *trans*-activating potential of IRF3 after its activation and the subsequent assembly of the multi-protein complex at the IFNβ enhancer that controls gene expression. Further, we will inspect the regulatory mechanisms of these steps imposed by the host cell and present the manifold strategies viruses have evolved to intervene with IFNβ transcription downstream of IRF3 activation in order to secure establishment of a productive infection.

## 1. Synopsis—IRF3 Activation Induces Transcription of *IFNB1* upon Viral Infection

### 1.1. The Name Says It All—Interferons Counter Infections

The interferon (IFN) system provides mammalian cells with a potent framework to fend off intruding pathogens. Interferons are signalling molecules first discovered more than 60 years ago, when virus-infected cells were found to release soluble compounds that could interfere with establishment of virus infection [1]. Since this initial discovery, we have come to understand the pivotal role of interferon signalling for the immune response to invading pathogens, from conveying the very first notice of intrusion to eliciting a well-tailored immune reaction suited to thwart the infection. Today, we differentiate three classes of interferons based on the receptor they employ for signal transduction. More than a dozen genes encoding IFNα subtypes and a single *IFNB1* gene give rise to the majority of type I IFNs in humans. They are the first messenger molecules released upon detection of a pathogen by infected cells and by bystanders to initiate the intrinsic defence mechanisms and to further involve dedicated cells of the immune system (recently reviewed in [2,3]). IFNγ, the only type II IFN, presents a further important regulator of the cellular immune defence mechanisms but is produced mainly by natural killer cells and T cells (reviewed in [4]). The biological activity of type III IFNs, represented by IFNλ1 to IFNλ4 and initially grouped to the interleukins, is confined to epithelial barriers where it balances infection control versus damaging inflammation (reviewed in [5,6]). Important abbreviations that are used throughout the text are listed in Appendix A.

### 1.2. The Setting—Cytosolic Nucleic Acids Stimulate Activation of Specific Transcription Factors

The production of type I IFNs is tightly regulated at multiple levels to enable a rapid induction on the one hand, but on the other hand to prevent overshooting or inadequate activity of these potent immune stimulators as this can lead to severe diseases. For instance, the multi-systemic disorder systemic lupus erythematosus is associated with dysregulation of the type I IFN system (reviewed in [7]). Essential to the induction of type I IFN transcription is the activation of two specific members of the interferon regulatory factor (IRF) family, a class of nine transcription factors (reviewed in [8]). IRF3 and IRF7 are activated upon the recognition of pathogen-associated molecular patterns (PAMPs) by pattern recognition receptors (PRRs). As viral molecules are produced by mammalian cells and do therefore not encompass structures inherently foreign to the host, detection of viruses greatly depends on recognition of nucleic acids. In fact, the immunostimulatory potential of DNA and RNA molecules was observed before the concept of PAMPs and PRRs was first mentioned [9,10,11]. Dedicated sensors distinguish self from non-self either based on a particular structure of the viral nucleic acids or due to a subcellular localization that does not occur in the healthy host cell (reviewed in [12]). In the first cells invaded during infection, usually fibroblasts, endothelial and epithelial cells, viruses are engaged by two classes of intracellular nucleic acid sensors when entering the cytosol: the retinoic acid-inducible gene I (RIG-I)-like receptors (RLRs) which sense aberrant RNA structures generated during cytosolic viral replication and several DNA sensors including cyclic GMP-AMP synthase (cGAS), interferon-gamma inducible protein 16 (IFI16) and DNA-dependent protein kinase (DNA-PK) that detect aberrantly localised double-stranded DNA (reviewed in [13]). In contrast, in immune cells like dendritic cells and macrophages that are specialized on surveillance by phagocytosis, sensing of nucleic acids mainly depends on the endosomal-localised Toll-like receptors (TLRs) TLR3, TLR7/8 and TLR9 (reviewed in [14]). Stimulation of all classes of PRRs is relayed by their adaptor proteins mitochondrial antiviral signalling protein (MAVS), stimulator of interferon genes (STING), myeloid differentiation primary response 88 (MyD88) or TIR domain-containing adaptor protein inducing interferon-beta (TRIF) to the activation of kinases which in turn phosphorylate and activate IRF3 and IRF7.

In parallel to IRF3/7, nucleic acid-mediated stimulation of the different signalling modules activates nuclear factor “kappa-light-chain-enhancer” of activated B-cells (NF-κB) by inducing dissociation from its inhibitory complex (reviewed in [15]) and activator protein 1 (AP-1) by mitogen-activated protein kinases (MAPKs) [16]. Along with IRF3/7 and coactivators, these transcription factors bind to regulatory DNA elements at the enhancer region upstream of the IFNβ promoter to induce gene expression (reviewed last in [17]). The interactions at the IFNβ enhancer leading to the formation of a multi-protein complex termed IFNβ enhanceosome are precisely concerted to allow a highly specific response. Here, we want to revisit the steps activating the *trans*-activation potential of IRF3, focusing on the subsequent interactions of this essential factor up to the formation of the enhanceosome that prompts the immediate initiation of type I IFN secretion upon virus detection. Further, we will point out the numerous mechanisms positioned by the host cell to control the type I IFN response and the sophisticated strategies deployed by viruses to evade it.

### 1.3. Meet the Lead—IRF3 and IRF7, the “Masters” of Type I IFN Transcription

Uniquely among the IRF family members participating in the type I IFN response, the IRF3 protein is constantly expressed in host tissues [18]. As depicted in Figure 1, the major human IRF3 isoform of 47 kDa and a length of 427 amino acids (AA) contains a highly conserved N-terminal DNA-binding domain (DBD) encompassing a helix-turn-helix signature with five well conserved tryptophan residues and a C-terminal IRF association domain (IAD) that mediates protein–protein interactions with other IRFs or coactivators (reviewed in [8]). These two major domains are connected by a flexible linker that contains a proline-rich segment [19]. A serine-rich auto-inhibition element (AIE) formed by the very C-terminus, also termed signal-response domain, and an AIE formed by a part of the linker keep IRF3 in an inactive state in resting conditions [19,20]. Phosphorylation at specific residues by stimulus-activated kinases induces conformational changes that enable IRF3 to interact with other proteins (see Section 2.1). The protein further contains a nuclear localization signal (NLS) formed by two basic clusters (K77/R78 and R86/K87) within the DBD, and a nuclear exit signal (NES) in form of a leucine-rich sequence between residues 139 and 149 [21,22,23]. Both NLS and NES are constitutively active and important for the function of IRF3 as discussed below (see Section 2.3). For simplicity, the above described variant 1 of the IRF3 isoforms will be referred to as IRF3.

IRF3 and IRF7 are the most closely related members of the IRF family with especially high conservation of the AIE, enabling formation of functional heterodimers and providing the basis for the shared activation mechanisms as well as the similar mode of action [24,25,26]. Only in plasmacytoid dendritic cells and macrophages, IRF7 is continuously expressed at high levels and crucial for the control of type I IFN induction following TLR7 and TLR9 engagement [27]. Since these immune cells produce the major share of type I IFNs in an infected host organism, IRF7 was termed the “master regulator” of IFN expression (reviewed in [28,29]). In most other cell types, however, IRF7 is expressed at very low levels in absence of stimulation [30]. In contrast to the ubiquitous IRF3, participation of IRF7 molecules seems less relevant during the very first response to virus detection at initial viral entry sites, which are usually composed of epithelial or endothelial cells or fibroblasts. 

Nevertheless, IRF7 is an important contributor of the type I IFN response. Upon IFNα/β signalling via the interferon-alpha/beta receptor (IFNAR), IRF7 expression is highly induced as part of the positive feedback regulation of the antiviral response; in other terms, IRF7 is the product of an IFN-stimulated gene (ISG) [31]. Newly synthesized IRF7 undergoes an activation similar to IRF3 and in concert with IRF3 further amplifies transcription of IFNβ. This behaviour can interfere with studies focused on IRF3 activity or modulation thereof in later stages. IRF3 is degraded in the course of its activity, as discussed below (see Section 4.3), while IRF7, though quickly degraded, is constantly expressed during stimulation [28]. This shifts the control of gene expression in fibroblasts, epithelial and endothelial cells from IRF3-mediated regulation upon initial sensing of an infection towards IRF7-mediated in later phases [32,33,34]. Knockout experiments in mice have shown that lack of IRF3 delays the immune response, while cells lacking IRF7 respond early but are unable to fend off the infection without the signal amplification [27]. Accordingly, together with the hefty feed-forward amplification of IFNα/β signalling, this feature is essential for the induction of distinct and diverse cytokine subsets by IRF7, especially of IFNα, that differentiate the reaction and prime cellular immunity [28,35]. For this reason, both IRF3 and IRF7 are crucial for the rapid induction and potent establishment of the antiviral response [27,34].

## 2. Preparation Is Everything—The Key Steps Enabling the Biological Role of IRF3 

The crucial role of IRF3 and the posttranslational changes it undergoes upon viral infection were first reported more than 20 years ago: Upon stimulation, IRF3 gets phosphorylated and accumulates in the nucleus where it interacts with the coactivators CREB-binding protein (CBP)/p300 to specifically bind to virus-inducible enhancer elements and exerts transcriptional activation of the *IFNB1* gene [21,36,37,38] (Figure 2). Since these first observations, our knowledge of the mechanism of action of IRF3 has been greatly refined. 

### 2.1. Dress up—Phosphorylation of IRF3 Enables Protein–Protein Interactions

Hyperphosphorylation is the first step in the activation of IRF3 as a functional transcription factor. In unstimulated cells, the IRF3 protein exists as two forms, an unphosphorylated (I) and a basally phosphorylated (II) form [39]. After viral infection, IκB kinase-epsilon (IKKε) and TANK-binding kinase 1 (TBK1), two homologs of the inhibitor of NF-κB kinase (IKK), are activated when PRRs convey the sensing of virus infection to their adaptor proteins, inducing conformational changes and interactions that lead to the formation of a new interaction surface (reviewed in [13]). The kinase binds to this signal-induced adaptor platform and phosphorylates the pLxIS motif (p: hydrophilic, x: any AA) on the surface of the adaptor proteins [40,41]. IRF3 is then recruited to the platform via its recognition site for the phosphorylated pLxIS motif and gets further phosphorylated, giving rise to two more protein forms (III and IV) whose appearance correlates with CBP interaction and IFNβ induction [39,42,43,44].

The serine- and threonine-rich region within the C-terminal AIE of IRF3 contains two clusters of potential phosphoacceptor residues: cluster 1 (S385/S386) and cluster 2 (S396/S398-S402/T404/S405) [19]. First, TBK1 phosphorylates cluster 2 residues of monomeric IRF3, and the additional negative charge induces a reorientation of the AIEs N- and C-terminal of the IAD [20,45]. This unmasks a hydrophobic binding pocket required for further protein interactions and renders the C-terminal tail (CTT) accessible for interactions [45,46]. Additionally, the conformational change is relayed to the DBD to enhance the DNA-binding affinity. The residues of cluster 2 are functionally largely redundant in terms of phosphorylation-mediated IRF3 activation, though phosphorylation of S396 seems to predominate in vivo [37,47]. The alternative use of phosphorylation sites could be the reason why some studies found that mutation of IRF3 S396 to alanine sometimes retained biological function (for example in [48,49]). Second, and induced by the first modification, IRF3 gets further phosphorylated at S386 which is pivotal for dimerisation [45,50]. Consistently, 100% of IRF3 dimers generated in vitro by incubation with TBK1 are phosphorylated at both clusters [45]. In contrast to cluster 2, added phosphate groups at cluster 1 residues have distinct effects: Addition of a phosphate group at S386 promotes dimerisation of IRF3 and strengthens interaction with the coactivator CBP, while phosphorylation of S385 negatively affects both interactions [19,50,51]. By relief of the auto-inhibition, phosphorylated IRF3 can dimerise and associate with the coactivators CBP/p300 to form an active holocomplex. IRF3 can also be rendered constitutively active by exchange of the five phosphoacceptor sites in cluster 2 to aspartic or glutamic acid residues (IRF3-5D or -5E, respectively), and these mutants display a strong tendency to acquire the cluster 1 phosphorylation and dimerise [19,45]. 

### 2.2. Join Forces—Dimerisation and Interaction with Coactivators Is Required for DNA-Binding of IRF3

Since the first description of IRF3 dimerisation, it is generally noted as the second step of activation after phosphorylation [19]. Formation of dimers requires homotypic interactions of the IAD with a second phosphorylated IRF3 molecule. Structural studies revealed that phosphorylation of IRF3 in fact modifies a pLxIS motif in the CTT, similar to the motif initiating recruitment of IRF3 to the adaptor platform [42]. Enabled by the negative charge at S386 after phosphorylation, the extended CTT can interact with the pLxIS-binding surface of a second phosphorylated IRF3 monomer to form a domain-swapped dimer (indicated in Figure 2). Zhao and colleagues further proposed that IRF3 dimerises at the adaptor platform to regain stability, but this was not yet confirmed. Due to repulsion caused by the newly acquired negative charges, IRF3 proteins could also dissociate from the adaptor complex before dimerisation and converge subsequently either (i) in the cytoplasm, (ii) after translocation into the nucleus or (iii) after engagement of coactivators during recruitment to the enhancer.

Further, the phosphorylation-induced rearrangement of the AIE unmasks a hydrophobic binding site of IRF3 and enables the interaction with transcriptional coactivators [21,37,52,53]. Association of IRF3 with the histone-modifying lysine acetyltransferases (KATs) CREB-binding protein (CBP, or KAT3A) and/or p300 (KAT3B) in form of a holocomplex is pivotal for the ability of IRF3 to *trans*-activate *IFNB1* expression [22,54,55]. These coactivators are large proteins that contain several folded domains and additionally a big share of intrinsically disordered regions that allow for specific binding to numerous factors upon interaction [56]. Their flexible structure enables the regulation of gene transcription by integrating the cues from several hundred transcription factors, yielding CBP/p300 the designation “master” coactivators of transcription. The closely related proteins CBP and p300 are usually regarded as functionally redundant—thus often referred to in combination as CBP/p300—and due to this assumption, studies characterising IRF3 interactions assessed association of one or the other [57]. However, there is growing evidence that their activity is overlapping but can be distinctly involved in regulation of different pathways, as exemplified by brain development [58,59]. Considering that both CBP and p300 can participate in holocomplex formation with IRF3 but do so with different shares [38,52], it would be interesting to determine whether engagement of CBP versus p300 is specific or coincidental. Potentially, the contribution of the individual coactivators could change during the different phases of *IFNB1* expression depending on integrated signals.

### 2.3. Enter the Final Scene—Protein–Protein Interactions Retain IRF3 in the Nucleus after Stimulation

To exercise its function as a transcription factor, it is pivotal that IRF3 can reach the nucleus. In fact, the latent protein already shuttles between the cytoplasm and nucleus in unstimulated conditions driven by its NLS and NES. As the influence of the NES seems to prevail the import, the main share of IRF3 molecules localises to the cytoplasm [22]. The constitutive export of IRF3 was shown to involve exportin 1 (CRM1) [21,22], and the import through nuclear pore complexes involves the importins karyopherin (KPN) subunit α2 (KPNA2), KPNA3 and KPNA4 (or Qip1) [22,60]. Upon infection, phosphorylation of the AIE and accompanying structural rearrangements enable IRF3 to associate with CBP, and this interaction retains the activated holocomplex in the nucleus [22]. 

Interestingly, in vitro experiments from several groups analysing the interaction between IRF3 and the IRF-binding domain (IBiD) in the C-terminus of CBP have shown that independently of dimerisation, monomeric IRF3 can interact with CBP once the auto-inhibition is relieved [22,45,46,51]. Moreover, in vitro analyses by Chen and colleagues implied that the presence of CBP promotes oligomerisation of IRF3 molecules with the required phosphomimetic mutations that enable dimerisation, while the mutant proteins stayed monomeric without CBP [51]. In a cellular scenario wherein IRF3 dimerises directly after acquiring the required modifications in the vicinity of the adaptor platform in the cytoplasm, this interaction might not be relevant, though. Characterisation of the holocomplex demonstrated that the homodimer of IRF3 is more stable than the holocomplex of IRF3 dimer plus p300 [53], and the association of IRF3 dimers with CBP is stronger than that of monomeric IRF3 [51], favouring a model wherein IRF3 first dimerises, and dimers subsequently interact with CBP. However, without determination of the precise temporal succession of interactions, this leaves the possibility that a phosphorylated IRF3 monomer enters the nucleus and interacts with CBP/p300 before it encounters a second IRF3 molecule to dimerise and swiftly induce a response, as suggested before [38]. In the end, the coactivator needs to associate with dimeric IRF3 because only this can induce activation of the KAT activity that is later required for assembly of the pre-initiation complex, as demonstrated for p300 [61].

### 2.4. Leave Room for Improvement—IRF3 Activity Comes in Different Ranges

Several groups noted exceptions to the common model of IRF3 activation followed above, drawing attention to the fact that assessment of the conventional signs like dimerisation that reflect the intermediate states is not always sufficient to infer on the downstream biological response [48,62,63]. Due to these discrepancies we suggest to differentiate between phosphorylated IRF3 that is relieved of the auto-inhibition and can “actively” participate in subsequent steps and IRF3 in the holocomplex that can exercise its *trans*-activation potential as a transcription factor to “actively” induce *IFNB1* expression. In fact, some of these studies assessed the activity of IRF3 in terms of induction of ISG56 [48,62], which intermingles the IRF3-dependent regulation of type I IFNs with directly IRF3-regulated ISG expression [64]. To our knowledge, it remains to be determined whether the form of IRF3 induced for up-regulation of specific ISG promoters is the same as the form of IRF3 able to induce *IFNB1* expression. 

Moreover, reports clearly reflect that there are different extents of IRF3 activity, and the protein might not yet be fully activated when initiating a first wave of *IFNB1* expression. Instead, IRF3 might acquire modifications and engage in distinct interactions in different waves of the type I IFN response until it is fully activated. In line with this, we noticed several reports referring to “full” activity of IRF3 in later loops of the response, for example enabled by phosphorylation at both clusters of the AIE or determined only in the presence of IFNAR signalling [55,65]. This aligns with the kinetics of the IFNβ response we will briefly revisit below (see Section 3.3), wherein only a part of the *IFNB1* alleles are engaged in the early response.

## 3. A Complex Performance—Stimulus-Induced Assembly of the IFNβ Enhanceosome

Expression of *IFNB1* is probably the best studied model of induced eukaryotic gene regulation. It exemplifies how factors activated by specific signals cooperate to prompt a defined expression programme, as reviewed before [17,66,67,68]. Upon viral stimulation, a multicomponent complex of transcription factors, coactivators and architectural proteins forms at the enhancer sequence upstream of the *IFNB1* gene promoter. This higher-order nucleoprotein complex, termed the IFNβ enhanceosome, works as a molecular switch that turns on upon virus-induced activation of a specific set of transcription factors [69,70]. When completed, the enhanceosome in turn recruits factors with chromatin-modifying activities as well as basal eukaryotic transcription factors to the nearby transcription start site (TSS) to initiate gene expression. The assembly is regulated at several levels and highly specific due to an elaborate organisation of transcription factor binding motifs, the required set of activated transcription factors and architectural proteins and finely coordinated interactions of the individual participants as well as of the arising composite structures.

### 3.1. The Preface—The DNA Blueprint for Inducible IFNB1 Expression

From just slightly upstream of the TSS of the human *IFNB1* gene, a precisely arranged sequence of *cis*-acting DNA constitutes the key enhancer element that defines *IFNB1* expression. The IFNβ enhancer itself is accessible yet directly flanked by two nucleosomes, one of them covering the TSS and a TATA box [71]. This setup strategically restricts the access of the transcription machinery in latent conditions [72,73]. The enhancer can be divided into four positive regulatory domains (PRDs) in the order IV-III-I-II that are discussed in detail in [68]. Of note, however, is the overall architecture that is essential for the precise recognition and following interplay of the array of core transcription factors [74]. 

The PRD farthest away from the promoter, PRDIV, features motifs for binding an AP-1 heterodimer formed by activating transcription factor 2 (ATF2) and c-Jun of the basic-region leucine zipper (bZIP) family [66,75]. Binding of the heterodimer over homodimers is favoured by the intrinsic architecture of the enhancer and important for subsequent interactions [76]. The last PRD in the sequence of the enhancer, PRDII, is recognised by a p50-p65 or p50-p50 NF-κB dimer [77,78,79]. PRDIII and PRDI in the middle section feature in total four overlapping IRF-binding elements (IRF-Es) of the consensus sequence 5′-AANNGAAA-3′ that form two composite binding sites enabling binding of two IRF dimers [34,36,80,81]. This region was also termed P31 because these two PRDs act as a functional unit, an enhanson [38]. Recent in-depth analysis of the binding requirements of different IRF members suggests that, in addition to a slight variation of the core GAAA recognition motif, the flanking regions along with the spacing in between the IRF-Es contribute to specific binding of distinct IRF family members, here favouring recruitment of IRF3 and IRF7 [81,82]. The nucleotide sequence of the IFNβ enhancer causes a bent conformation of the DNA helix in inactive conditions and in this way drives cooperative binding of the transcription factor heterodimers upon stimulation [68].

### 3.2. On the Marks—Assembly of the IFNβ Enhanceosome 

After stimulus-induced activation, the transcription factors translocate into the nucleus. First, the NF-κB dimer p50-p65 is recruited and nucleates assembly [71]. Association of NF-κB is the most limiting factor of enhanceosome assembly, and to optimise recruitment, it is initially targeted to three loci at distinct chromatin regions via Alu elements, also termed NF-κB reception centres (NRCs) [71,83,84]. The transcription factor ThPOK (T-helper-inducing POZ/Krüppel-like factor, also ZBTB7B) binds cooperatively with NF-κB to these regions, and oligomerisation of ThPOK mediates interchromosomal interactions of the NRCs with the IFNβ enhancer to deliver NF-κB [83,84]. Alongside NF-κB, the architectural DNA-binding high mobility group protein I(Y) (HMG I(Y), also HMGA1) binds at two sites to the minor groove in PRDII and straightens the DNA conformation, thereby supporting interaction of NF-κB [69,85,86,87,88]. Subsequently, the remaining virus-induced transcription factors are recruited. A second HMG I(Y) molecule interacts with ATF2-c-Jun to promote binding of the heterodimer to PRDV by straightening the DNA helix [69,88,89]. Additionally, NF-κB mediates capture and binding of the IRF3-CBP/p300 holocomplex [38,90]. In vitro, IRF3 displays a surprisingly weak affinity for P31, which led to the former model in which IRF1 is the critical IRF member in *IFNB1* induction because it freely associates with the enhancer sequence [91,92,93]. However, IRF1 was then found to be dispensable for *IFNB1* expression upon PRR activation, which is congruent with its expression pattern as an ISG [94,95,96]. Instead, when phosphorylation releases the auto-inhibition of IRF3 and enables interaction with CBP/p300, the coactivator strongly increases the affinity of IRF3 for DNA and enables association of the complex [18,38,54,55]. Further cooperativity is mediated by the P31 sequence that ensures concerted binding of ATF2-c-Jun and IRF3 [97,98] and by CBP/p300-mediated interactions within the enhanceosome complex [38,90]. Clustering of IRF3 target regions with binding motifs for NF-κB or other transcription factors along with low chromatin accessibility in steady-state conditions was recently described as a common feature of IRF3-DNA interactions, implying an obligatory collaborative binding mode of IRF3 that promotes access to chromatin [82]. At the IFNβ enhancer, a total of four IRF3/7 molecules bound to four IRF-Es in tandem are required to induce gene expression of a luciferase-based IFNβ-reporter plasmid in the commonly used human embryonic kidney cell line HEK293T [99]. Structural analysis suggests that one IRF dimer binds on one side of the duplex, occupying the first and the third IRF-E, and the second dimer binds from the opposite side, binding to the second and fourth IRF-E [81]. 

Remarkably, the DBDs of the eight transcription factors barely interact with each other despite the great overlap of their binding elements [81]. Instead, the high level of cooperativity between binding events is achieved by the specific order of individual response elements that allows for binding-induced conformational changes of the DNA to transmit allosteric effects [68,74,81,100]. Molecular dynamics simulations by Pan and Nussinov further showed that the combination of overlapping recognition elements with an alternation between consensus and non-consensus motifs optimises the specific interactions by enhancing or restricting binding of neighbouring transcription factors. Moreover, the signal integration by CBP/p300 that interacts with each transcription factor through different domains is essential for the functionality of the enhanceosome [90]. In addition to the pivotal architectural role of recruiting IRF3 and mediating interactions with the transcription machinery as coactivator, CBP/p300 also participates in the subsequent chromatin remodelling in its vicinity [71,101,102]. As implied above, however, the precise number and identity of coactivator molecules participating in the formation of the active IRF3-CBP/p300 holocomplex as well as in the enhanceosome is unknown. Potentially, they present a module for further signal integration by participating in dynamic compositions. Further, also the subtle modulation by the two HMG I(Y) proteins is required for the maximal level of expression [86], though their participation in the final assembly is controversial as previously discussed [17]. In line with the high synergy during assembly, the IFNβ enhanceosome is extraordinarily stable and enables multiple rounds of re-initiation in vitro [70,88]. Due to the binding-induced conformational changes of the DNA foundation, the fully assembled structure encompasses a straight segment of DNA covered by eight specifically bound transcription factors all of which interact with the essential coactivator CBP/p300. This final IFNβ enhanceosome presents a single new, continuous activating surface and provides the basis for the association of the basal transcription apparatus.

### 3.3. Get Ready and Go—Initiation of IFNB1 Transcription

Before transcription of the *IFNB1* gene can commence, the chromatin landscape needs to be modified so that the pre-initiation complex can be assembled and access the TSS. First, the IFNβ enhanceosome mediates the transient recruitment of a complex of p300/CBP-associated factor (PCAF, or KAT2A) and general control nonderepressible-5 (GCN5 or, KAT2B) to the promoter and induces the remodelling, starting with GCN5-mediated acetylation of nucleosomes and HMG I(Y) in its vicinity [70,71,103]. Acetylation of HMG I(Y) at K71 by GCN5 strengthens the association of NF-κB and thereby further stabilises the enhanceosome [70,71]. Next, the modification of the histone tails of nucleosomes at H3 and H4 promotes recruitment of the chromatin remodelling BRG1- or BRM-associated factor (BAF) complex of the SWI/SFN family [71,87,90,102,103]. In parallel, general eukaryotic transcription factors (TFs) TFIIA, TFIIB, TFIIE, TFIIH, TFIIF and upstream stimulatory activity (USA) cofactors assemble at the promoter along with the RNA polymerase II (RNAP II) holoenzyme [71,87,101]. Stimulated by the environment rich in acetylated histones, the ATPase Brahma-related gene 1 (BRG1 or SMARCA4) of the BAF complex induces a conformational change of the nucleosome at the TSS [71,104]. Now, the TATA box at the TSS is available for binding by the TATA box-binding protein (TBP) and allows association of the TFIID complex [71,103]. Notably, TFIID binds here after the RNAP II complex, as opposed to the classical sequence of events in assembly of the pre-initiation complex [105]. Association of TFIID induces a bend of the DNA helix so that the nucleosome covering the TATA box slides 36 bp downstream of its latent position [73,106]. With this, the arrangement of the pre-initiation complex can be completed, and *IFNB1* transcription can begin [17,107]. 

Notably, while the rapid induction of the type I IFN response is crucial for the host defence, the first round of *IFNB1* expression yields only low levels of IFNβ, as discussed by Ford and Thanos [17]. Briefly, after stimulation of a cell population, *IFNB1* is initially expressed only by a fraction of cells and from only one allele due to a stochastic phenomenon that indicates a limiting amount of required cellular factors [108,109,110]. The initial recruitment of NF-κB to other loci that collect the available transcription factor and transfer it to the required binding sequence highlights the p50-p65 heterodimer as the most limiting factor [83,84]. The weak initial signal of secreted IFNβ induces high-level expression of IRF7, and newly synthesized IRF7 promotes enhanceosome assembly and *IFNB1* transcription from the remaining allele and in more cells [83]. Finally, the weak initial IFNβ signal is then amplified to eventually induce the cytokine storm of the antiviral response.

### 3.4. The Run-Through—Basal Transcription of IFNB1

The expression of *IFNB1* is, at least in murine cells, tightly regulated by different forms of NF-κB dimers in the initial response to infection, as reviewed by Balachandran and Beg [111]. At resting conditions, p50 homodimers associate with the IFNβ promoter region and repress basal transcription in the absence of appropriate stimulation [66,112,113]. In addition, a binding site for NF-κB regulatory factor (NRF) overlaps the NF-κB binding element in PRDII and negatively affects basal transcription [114,115]. In line with a primary inhibitory effect, knockout of murine p50 does not affect induction of *IFNB1* expression after virus infection [116]. However, the p50-imposed down-regulation in resting cells is competed with by p50-p65 dimers that are constitutively activated by IKKβ and cycle between the cytoplasm and nucleus [117]. This low level activity of p65 supports a basal level of *IFNB1* expression that ensures rapid and robust responsiveness on demand, while the negative regulation by p50 ensures specificity [113]. After stimulation, p50-p65 is required to assist in the early recruitment of IRF3 together with CBP/p300 to support *IFNB1* expression [90,113,118]. Only at high levels of IRF3 activity, i.e., at the peak of activity later in infection or brought about by expression of the constitutively active IRF3-S396D, p50-p65 is dispensable for *IFNB1* transcription [118]. Subsequent to the initial recruitment, the increasing levels of active IRF3 and IRF7 can fully assume the regulation of IFNβ production, and NF-κB proceeds to regulate expression of pro-inflammatory genes [118]. The p50-p65-mediated basal expression of *IFNB1* raises the question how recruitment of the transcription machinery is achieved at resting conditions, considering that IRF3 is missing to participate in a highly cooperative assembly as described above. Moreover, since the type I IFN system is not 100% conserved between mice and humans, and all of these studies were carried out in murine systems, it remains to be confirmed whether the exact same mode applies for the participation of NF-κB in the human system. 

### 3.5. Backstage—Long-Range Modulation of IFNB1 Expression

In addition to the proximal enhancer, DNA elements further away from the promoter are involved in *IFNB1* transcription, though again, most studies have thus far focused on the murine system. In non-infected mouse cells, a proximal region of the *IFNB1* locus that contains binding sites for the transcription factor Yin Yang 1 (YY1) was shown to mediate association with pericentromeric heterochromatin (PCH), a feature related to gene silencing [119]. After infection, the *IFNB1* locus is repositioned away from PCH, and this observation correlated with transcriptional activation of the promoter shortly thereafter. Josse and colleagues suggested that this effect could be mediated by binding of YY1 to the proximal promoter. Furthermore, YY1 interacts also in the absence of viral infection in mice with the signal transducer and activator of transcription 1 (STAT1), one of the main factors mediating ISG expression [35], implying important roles of YY proteins at different stages during infection [120]. In the human cell, YY1 and YY2 regulate enhanceosome formation from YY-binding site-containing regions far upstream of the TSS of the *IFNB1* gene at −3 kb and −2 kb, termed DNase I hypersensitive sites 1 (HS1) and HS2, respectively [121]. YY1 and YY2 seem to modulate each other, with YY2 antagonising the negative effect of YY1. In analogy to the murine IFNβ promoter region, this long-range interaction was suggested as requirement for the recruitment of GCN5 to initiate chromatin remodelling at the IFNβ promoter [121,122,123]. 

A further DNA element regulating *IFNB1* transcription is the long-range enhancer L2, which was first identified in human cells but is conserved in mice [124]. L2 is activated upon viral stimulation and recruits IRF3 and RNAP II dependent on the IRF-E within its interferon-stimulated response element (ISRE). This induces expression of a non-coding RNA from the enhancer, and this enhancer RNA (eRNA) in turn supports *IFNB1* expression. However, the mode of action of the eRNA was not yet determined.

## 4. All in Moderation—The Many Ways to Adjust IRF3 Activity

The rapid induction of the antiviral state is critical to overcome a nascent infection. While an insufficient response would be beneficial for the pathogen, overshooting activation of the immune system can cause severe damage to the organism. Therefore, several mechanisms are implemented by the host organism to regulate IRF3 at all levels, from moderating the potential to activate IRF3 from its latent state, to promoting its participation in formation of the IFNβ enhanceosome, to supporting its activity during ongoing stimulation, through to terminating the response with deliberate timing. Factors that contribute to the finely tuned ensemble of modulators can be continuously active, induced or inhibited upon stimulation. A wide array of molecular mechanisms that modulate the biological activity of IRF3 are deployed, including protein–protein interactions, regulatory RNAs and common as well as non-canonical posttranslational modifications. To illustrate the abundance of strategies, we will include also proteins expressed predominantly in immune cells. In particular, mechanisms aiming at dampening the activity of IRF3 were usually identified in macrophages, highlighting the importance to keep the response primed but low until the full activation becomes necessary. Since posttranslational modifications of IRF3 and especially phosphorylation and ubiquitination were reviewed before [125,126], we will instead focus on how the host factors engage in the different steps to modulate the biologic activity of IRF3 (Figure 3).

### 4.1. Support for the Key Actor—Host Factors that Promote IRF3 Activity after Viral Stimulation 

In addition to the essential steps of IRF3 activation delineated above, various protein–protein interactions and additional posttranslational modifications were demonstrated to support the biological activity after induction. Before stimulation, for example, protein phosphatase 1 (PP1) removes the phosphate groups at S385 and S396 in macrophages and consequently attenuates IFNβ production [127]. Early after TLR- or RLR-mediated stimulation, however, activity of PP1 is down-regulated to increase the pool of active IRF3 and augment the immune response [127]. Also first identified in murine macrophages, the lysine methyltransferase nuclear receptor-binding SET domain 3 (NSD3) targets nuclear IRF3 after viral stimulation and modifies IRF3 by adding a single methyl group at K366 [128]. This modification promotes the dissociation of phosphorylated IRF3 from an isoform of PP1 catalytic subunit γ, thereby hindering the inhibitory effect of PP1 and maintaining the activated state of phosphorylated IRF3. Similarly, type I IFN up-regulates the expression of the long non-coding RNA lncLrrc55-AS, an antisense transcript to the gene of leucine-rich repeat-containing protein 55, which supports IRF3 phosphorylation in macrophages by inhibiting an inhibitor [129]. In the cytosol, lncLrrc55-AS associates with the protein phosphatase methylesterase 1 (PME-1), which in turn promotes the interaction of PME-1 and the protein phosphatase 2A (PP2A). PP2A is an inhibitor of IRF3 signalling, but lncLrrc55-AS mediates its demethylation and inactivation to maintain IRF3 activity [129,130,131]. Additionally, stability of the IRF3 protein is promoted within the loop of the type I IFN response by covalent conjugation of newly synthesised ISG15 to IRF3 on K193, K360 or K366 mediated by Herc5 (HECT and RLD domain-containing E3 ubiquitin protein ligase 5) [132,133]. This modification, termed ISGylation, disturbs the interaction between IRF3 and peptidyl-prolyl isomerase 1 (Pin1), delays the proteolytic degradation of IRF3 and in this way prolongs the immune response [132,133,134]. The dual use of IRF3 K366 for methylation by NSD3 versus ISGylation by Herc5, both of which contribute to protein stability, hints at a complex network that also modulates the modulators.

While inactive IRF3 already shuttles between the cytosol and nucleus in latent conditions, the translocation becomes crucial after its activation in order to exercise the *trans*-activation activity. Just recently, Cai and colleagues reported that the ubiquitin specific peptidase 22 (USP22) promotes the antiviral response from the cytoplasm by deubiquitinating importin KPNA2 [60]. After viral infection, KPNA2 associates with IRF3 for nuclear import, and USP22 promotes this critical step by stabilizing KPNA2. Importin-mediated translocation is additionally regulated by the widely expressed transcription regulators Yes associated protein 1 (YAP1), YAP2 and YAP4: YAP1/2/4 associate with latent as well as activated IRF3 and block further interactions required for dimerisation or nuclear import [135]. After virus infection, however, activated IKKε phosphorylates a conserved motif of the YAPs, which triggers their lysosomal degradation and reliefs the YAP-mediated inhibition. Further, an additional phosphate group within the DBD of IRF3 at S97 is important for the nuclear translocation after viral stimulation [136]. This modification inhibits the nuclear import, and mass spectrometry revealed that about a fifth (18.2%) of IRF3 molecules carry it at latent conditions when exogenously expressed in HEK293. Upon viral stimulation, the dual-specificity protein phosphatase and tensin homolog (PTEN) removes this phosphate group and consequently the negative regulation of IRF3 [136]. Finally, to keep IRF3 in the nucleus, DNA-PK is activated in response to virus infection and phosphorylates IRF3 at T135 [137]. This phosphate moiety mediates the nuclear retention and delays proteolysis of the active transcription factor, thereby prolonging the IRF3-driven response. 

The association of CBP and IRF3 in the nucleus is supported by the constitutively active enzyme glutaredoxin-1 (GLRX or GRX1), which acts in the cytoplasm [138]. In resting cells, inactive IRF3 is S-glutathionylated and this modification would impede the essential interaction of IRF3 and CBP/p300. However, GLRX removes this modification after infection and thus supports IRF3 activity. While the deglutathionylation of IRF3 is independent of its phosphorylation or dimerisation state, the accompanying structural changes could be involved in the recruitment of GLRX to remove the modification after initial activation events [138]. Within the nucleus, the interaction of IRF3 and CBP is further promoted by a ubiquitously expressed subunit of the endosomal sorting complex required for transport (ESCRT)-II [139]. After virus infection, a fraction of the ESCRT-II subunit ELL-associated protein of 30 kDa (EAP30, also known as SFN8) localises to the nucleus where it interacts with activated IRF3 and CBP and promotes binding of the holocomplex to target gene promoters. Contrasting this, the protein Argonaut 2 (AGO2), a component of the RNA-induced silencing complex, interacts with the IAD of IRF3 and inhibits its association with CBP in the nucleus in latent conditions [140]. Upon viral infection, however, AGO2 is exported from the nucleus, and its inhibition is revoked to promote IFNβ production during infection [140]. 

A further mechanism of signalling amplification is set into motion by type I IFN-stimulated production of the E74 like ETS transcription factor 4 (ELF4). ELF4 is recruited to STING and activated by TBK1 similar to IRF3 after viral stimulation, though without affecting IRF3 activation itself [141]. Activated ELF4 enters the nucleus and binds to the IFNβ promoter region via ETS/IRF composite binding elements (EICEs). DNA binding of ELF4 synergises with binding of IRF3, and thus promotes enhanceosome formation. The critical role of ELF4 in the feed-forward amplification of the murine antiviral immune response was demonstrated in vivo, and notably, it is independent of the signalling in plasmacytoid dendritic cells [141]. In contrast, a mechanism that due to the predominating expression of both factors in tissues of the immune system applies exclusively to immune cells is the association of IRF8 together with the transcription factor PU.1 (also SPI1, Spi-1 proto-oncogene) to the IFNβ promoter region mediated by EICE [142]. Li and colleagues proposed that IRF8 and PU.1 support the rapid induction of transcription by forming a scaffold at the IFNβ enhancer to facilitate recruitment of activated IRF3 [142].

### 4.2. Balance Is Key—Host Factors That Attenuate IRF3 Activity

A striking feature of IRF3 modulation is reflected by mechanisms that are installed not to plainly heighten or terminate the biological activity of IRF3 but to moderate its extent. Some of the mechanisms delineated here affect the latent protein, others specifically target activated IRF3 after viral stimulation and still others act on both. Macrophages are a prime example in deploying numerous modulators to dampen fluctuations of IRF3 activity until a certain threshold is passed and further to restrain the response once unleashed. For example, the ubiquitin-protein ligase E3C (UBE3C) mediates K48-linked ubiquitination of IRF3 and IRF7, thereby targeting the master transcription regulators in dendritic cells for proteolysis, irrespective of their activation state [143]. In this way, UBE3C helps to maintain low amounts of IFN production in resting conditions and additionally restrains the magnitude of the response after stimulation [125,143]. Another E3 ubiquitin ligase, tripartite motif-containing 21 (TRIM21 or Ro52), was described by two groups to be important for regulation of IRF3 but with opposing findings as already discussed by Sin [125]: Higgs and colleagues reported TRIM21-dependent mediation of IRF3 degradation 48 hours after infection [144], whereas Yang and colleagues found that nine hours after infection, TRIM21 interferes with the interaction between IRF3 and Pin1 and in this way prevents Pin1-mediated ubiquitination and degradation of IRF3 to sustain the immune response [145]. Sin suggested that, in addition to effects potentially imposed by different cell lines, the observation at different times of ongoing infection could have contributed to the contrasting findings [125]. 

Stability of IRF3 is further modified by addition or removal of small ubiquitin-like modifier (SUMO) proteins. In HEK293 cells, endogenous IRF3 is SUMOylated at K70 and K87 in its DBD, and viral infection slightly increases SUMOylation [146]. The widely expressed human sentrin/SUMO-specific protease 2 (SENP2) removes these SUMO moieties and in this way conditions IRF3 for K48-linked ubiquitination at the same residues and subsequent degradation, thereby dampening the antiviral response [146]. SENP2 targets wild-type IRF3 as well as the constitutively active IRF3-5D mutant, suggesting that the SUMOylation initially masks the protein from degradation to prolong its activity but that this modification is constantly countered by SENP2 activity. In addition, stimulus-induced SUMOylation of murine IRF3 at K152 was reported to attenuate the IFNβ response irrespective of the protein’s phosphorylation status, though the mediating proteins are unknown [147].

Regulation of auto-inhibition aside, phosphorylation of IRF3 can contribute to the down-regulation of its activity also at subsequent steps. When screening the human kinome for phosphorylation events during nucleic acid sensing, Meng and colleagues identified mammalian sterile 20-like kinase 1 (Mst1, also STK4) as an inhibitor of IFNβ promoter induction and confirmed the effect of Mst1 in knockout mice [148]. Mst1 suppresses activation of TBK1 and IKKε and thereby inhibits the phosphorylation-dependent activation of IRF3, but it also targets IRF3 directly for deactivation by phosphorylating two sites of IRF3 (T75, T253). Demonstrating that this effect is independent of the inhibition of the upstream kinases by Mst1, introduction of a single phosphomimetic mutation, T253D, is sufficient to impair the ability of the constitutively active IRF3-5D to dimerise after stimulation [148]. 

A further level of negative regulation of IFNβ signalling is imposed by the ubiquitously expressed Fas-associated factor 1 (FAF1), which interacts with importin 5 (IOP5, also KPNB3) in resting as well as stimulated cells [149]. After viral stimulation, IRF3 increasingly associates with IPO5 for nuclear import, but this is dampened by the FAF1-IPO5 interaction which consequently reduces the translocation of phosphorylated IRF3 and the induction of *IFNB1* expression. 

In the nucleus, formation of the IRF3/p300 complex and p300-mediated acetylation of IRF3 after virus infection is assisted by bromodomain-containing 3 (Brd3) [150]. Generally, interaction of Brd3 with p300 promotes recruitment of the holocomplex to the IFNβ enhancer and facilitates *IFNB1* transcription. This supporting mechanism seems to come with a time limit, however, as analysis in macrophages revealed that viral stimulation specifically down-regulates Brd3 abundance and thereby terminates its support [150]. Similarly, Zhang and colleagues recently reported stimulation-induced non-canonical K6-linked ubiquitination at three positions in the DBD of IRF3 (K39, K98, K105) and demonstrated that this modification is essential for the DNA-binding capacity of IRF3 in macrophages and in HEK293T cells [151]. As a counter measure, virus-infected cells additionally up-regulate expression of the ovarian tumour domain-containing deubiquitinase (OTUD1), and OTUD1 then removes this ubiquitination from IRF3, resulting in reduced IRF3-binding to the IFNβ promoter region at later times [151]. 

To allow moderation of enhanceosome assembly after formation of the active IRF3-CBP/p300 complex, other DNA-binding proteins were reported to interact with motifs contained within the IFNβ enhancer sequence. The first IRF-E overlaps with a consensus binding site for an activator of pro-inflammatory responses in macrophages, NFAT5 (nuclear factor of activated T cells 5), which is conserved between human and mouse promoter sequences [152]. NFAT5 constitutively forms dimers that can bind to the IFNβ enhancer region and competes with IRF3 due to the overlapping recognition sequence, resulting in limited recruitment of IRF3 to the enhancer and thus limited promoter induction. In a similar way, the transcription regulator MAF bZIP transcription factor B (MafB) binds in macrophages to the IFNβ enhancer sequence mediated by AP-1-like sites [153]. In resting conditions, MafB is a weak positive regulator of basal *IFNB1* transcription. Upon stimulation and activation of IRF3, however, it impairs the interaction between the coactivators and antagonises enhanceosome formation. This dual role was suggested to allow the system to deal with fluctuations in IRF3 activity [153].

### 4.3. The Curtain Drops—Host Factors that Terminate IRF3 Activity after Viral Stimulation

During the course of the innate immune response, the signal that started it all has to be switched off again to allow other messengers to refine the defence line. The first mechanism to terminate the IRF3-dependent response is set into action directly during the activation of IRF3 by phosphorylation of the C-terminal AIE [37]. In addition to releasing the auto-inhibition, the modification represents a signal for degradation of the protein, and so activated IRF3 has a shorter half-life than the latent form [37,154]. In this way, turnover of the activated protein by degradation rather than additional activation of repressors is an integral feature responsible for ending *IFNB1* expression [155]. Several molecules have been reported to mediate ubiquitination of IRF3 in order to induce its depletion after initial stimulation [126]: Pin1, FoxO1 [156], TRIM26, RBCK1 and TRIM21 (see above, Section 4.2) all mediate K48-linked ubiquitination and subsequent degradation of activated IRF3. For example, the peptidyl-prolyl isomerase Pin1 specifically mediates degradation of the activated transcription factor by recognition of a further phosphorylated motif (S339phos-P340) IRF3 acquired during stimulation [134]. Additionally, viral infection promotes the nuclear localisation of the E3 ubiquitin ligase TRIM26, allowing TRIM26 to bind phosphorylated IRF3 in the nucleus and promote its K48-linked ubiquitination and degradation [157]. Similarly, production of RBCC protein interacting with PKC1 (RBCK1), another E3 ubiquitin ligase, is induced by viral stimulation and targets IRF3 for ubiquitination [158]. Furthermore, caspase-8 (CASP8) is activated by cytosolic RIG-I-dependent signalling and cleaves IRF3 at a recognition motif between DBD and IAD (^118^SQPD^121^), inducing ubiquitination and degradation of the fragments [159]. 

In parallel to the activation of IRF3, viral stimulation also activates inhibitors of IRF3. The transcriptional regulator Krüppel-like factor 4 (KLF4), which is widely expressed in human tissues, increasingly localises to the nucleus after viral infection where it binds to the IFNβ promoter region [160]. This reduces recruitment of IRF3 and thus inhibits induction of *IFNB1* expression. In immune cells, the lysine acetyltransferase KAT8 was identified by an siRNA-screen as a negative regulator of antiviral innate immunity [161]. Viral infection promotes KAT8-mediated acetylation of IRF3 at K359, independent of the phosphorylation and dimerisation status of IRF3, and this modification interferes with binding of IRF3 to the IFNβ enhancer. A potential effect on assembly of the IRF3-CBP/p300 holocomplex was not characterised, however, leaving the exact step of interference to be determined.

A further line of down-regulation is introduced by the expression of ISGs that negatively modulate IRF3 activity. Considering the multitude of induced ISGs [162], it is not surprising that they target various levels to terminate *IFNB1* transcription. Nuclear import is repressed by the increasing interaction of IRF3 with the IFN-inducible DEAD-box RNA helicase DDX56, as this interaction competes with the association of IRF3 and IPO5 [163]. The IRF3-DNA interaction is inhibited by interaction of newly synthesized cell growth-regulating nucleolar protein LYAR (Ly1 antibody-reactive), which is usually low expressed in most cell tissues but induced by the IFNβ response [164]. LYAR interacts specifically with the N-terminal domain of activated IRF3 and in this way interferes with the DNA-binding capacity of IRF3 in the IRF3-CBP/p300 holocomplex. Additionally, a long non-coding RNA directly targets the IFNβ promoter region and interferes with transcription factor binding in a unique way [165]: After RNA deep sequencing revealed enhanced transcription of the long non-coding RNA lnc-MxA from the MxA locus after viral infection, Li and colleagues recently reported its mechanism of action. Applying a chromatin isolation by RNA purification assay and in vitro pulldown of an IFNβ promoter dsDNA fragment with biotin-labelled lnc-MxA, they demonstrated that lnc-MxA forms an RNA-DNA triplex with the IFNβ promoter region. This changes the structure of the chromatin and interferes with binding of the transcription factors IRF3 and NF-κB to their respective target sequence [165]. Finally, the delayed generation of PRDI-binding factor 1 (PRDI-BF1, or PR/SET domain 1) mediates recruitment of the histone H3 lysine methyltransferase G9a for epigenetic silencing of *IFNB1* expression [166,167].

The following host factors were additionally reported as negative modulators of IRF3 activity, but it remains to be seen if they constantly inhibit IRF3 or how they are regulated. For instance, calmodulin-like protein 6 (CALML6 or CAGLP) was identified to negatively modulate IRF3 and independently of the protein’s ability to bind calcium ions [168]. CALML6 interacts with the C-terminal AIE of IRF3, and this interaction is strengthened when IRF3 is phosphorylated. After virus infection, CALML6 thereby impairs dimerisation and nuclear import of IRF3. The contribution of cellular FLIP long isoform protein (cFLIP_L_) to regulation of the type I IFN response in several primary cancers was reported by several groups with contradicting outcomes, so Gates and colleagues studied the underlying mechanism of action and concluded on an inhibitory function [169]. When ectopically expressed in HEK293T cells, cFLIP_L_ interacts with phosphorylated IRF3 within the nucleus and hinders interaction with CBP and thus with the IFNβ enhancer. cFLIP_L_ was then confirmed to be highly expressed in several human cancer cell lines and to interact with endogenous IRF3, mediating reduction of ISG expression. 

As opposed to the major isoform (variant 1) of IRF3 that drives the type I IFN response as discussed so far, three of the five described splice-variants of human IRF3 negatively modulate the biological activity of IRF3, though their regulation remains to be determined. Translation of variant 2 (IRF3-CL) yields a slightly longer protein (452 AA) with a unique sequence of 125 AA in the C-terminus that lacks a part of the IAD including the AIE of IRF3 [170]. IRF3-CL constitutively forms homodimers that localize to the cytoplasm. After viral stimulation, IKK-mediated activation induces association of IRF3-CL with phosphorylated IRF3, but the heterodimers are retained in the cytoplasm and consequently, the association with this isoform keeps IRF3 from the induction of *IFNB1* expression. In contrast, variant 3 (IRF3a) lacks the N-terminal part of the functional DBD of IRF3 and is consequently unable to bind to classical IRF-Es [171,172]. This isoform also heterodimerises with IRF3 variant 1 after stimulation, but in line with the absent DBD, its presence selectively inhibits virus-induced *IFNB1* transcription. Interestingly, after virus stimulation, IRF3a is degraded slower than phosphorylated IRF3 and so the ratio of negative versus positive modulator increases with ongoing infection, potentially contributing to a downregulation of the type I IFN response [171]. The fourth variant, IRF3e or IRF3-nirs, was discovered when Marozin and colleagues searched for the reason of the defect in *IFNB1* expression in hepatocellular carcinoma (HCC) cells and discovered that a truncated variant of IRF3 was constitutively expressed in primary cells of HCC or HCC cell lines but not in primary hepatocytes [173]. IRF3-nirs is produced when an in-frame exon is aberrantly skipped during splicing, leading to the generation of a protein lacking 127 AA within the β-sandwich core of the IAD. In line with this, this isoform is constitutively active and maintains the DNA-binding ability. However, due to the compromised IAD, IRF3-nirs seems unable to exercise *trans*-activation activity and instead competes with IRF3 for the limited IRF binding sites in the IFNβ enhancer sequence [173]. Surprisingly, given that association with the coactivator is normally essential for IRF3 to bind to DNA motifs, this implies that the interaction surface for CBP/p300 remains intact despite the missing adjacent structural module. In addition to the human isoforms, one variant of murine IRF3, mIRF-3a, was reported as a ubiquitously expressed negative modulator of IFNβ induction in mice [174]. During the generation of mIRF-3a, an alternative donor splice site in exon 6 is used and leads to a frameshift with a premature termination codon, yielding a shorter variant (296 AA) which differs in the C-terminal region as compared to the major murine variant (419 AA). mIRF-3a freely localises to the nucleus, binds to IRF recognition sites and represses promoter activation.

## 5. Saboteurs of the Main Act—Viral Modulation of Activated IRF3

Not long after the initial steps in the characterisation of the IFNβ enhanceosome were made, also the first viral proteins targeting its assembly were noticed [175,176]. Today, a long list of viral factors targeting every step from the stimulation of PRRs by nucleic acids, to induction of transcription factor activation, to activity of IFN, through to the stimulation and activity of ISGs are known, and the list still grows [177,178,179,180,181,182,183,184]. The central role of IRF3 in this pathway makes it an attractive target for viral evasion. Strategies applied to interfere with the activation of IRF3 in the cytoplasm include (i) the inhibition of IRF3 expression; (ii) direct antagonism of essential phosphorylation events by targeting the essential kinases, their interaction with IRF3 or dephosphorylating IRF3 (recently joined by herpes simplex virus (HSV)-2 immediate early protein ICP27 [185], and the nucleocapsid protein of Peste des petits ruminants virus [186]); and (iii) mediating degradation of IRF3. Remarkably, even nuclear phosphorylated IRF3 can specifically be targeted for ubiquitination and proteasome-mediated degradation after activation [187]. Other viral factors aim to evade the type I IFN response more generally by limiting its induction, affecting production of IFN by transcriptional or translational shut-off or dysregulating the processing or trafficking of host mRNAs [188,189]. Noting that strategies to inhibit IRF3 activation and activity applied by viruses were frequently reviewed for specific virus families as well as in broad summaries [190,191,192,193,194,195], we want to emphasize here the molecular mechanisms viruses deploy to obstruct the function of IRF3 after its *trans*-activation potential is enabled by phosphorylation (Figure 4).

### 5.1. Dispersing the Winning Team—Inhibition of Dimerisation

As the activation of IRF3 is a multistep process, in early studies the exact level of viral interference sometimes remained elusive. Dimerisation of IRF3 was among the first steps reported to be targeted in order to antagonize its *trans*-activation activity. The ML protein, a splice variant of the matrix protein of Thogoto virus (THOV) of the family *Orthomyxoviridae,* blocks the formation of IRF3 homodimers irrespective of the presence of the C-terminal AIE and further inhibits association of IRF3 with CBP [196,197]. Interestingly, this modulator does not affect nuclear translocation, suggesting that ML does not disable import of IRF3 but renders otherwise activated IRF3 monomers unable to participate in further interactions in the nucleus. The authors ruled out a lasting interaction between ML and IRF3 that could for example directly compete with binding to CBP/p300 because they could not detect association of ML and IRF3 by co-immunoprecipitation [197]. Additionally, ML interferes with a later step in the induction of *IFNB1* expression, namely, the cofactor function of TFIIB at promoters that require de novo recruitment of RNAP II, which was indicated by the finding that ML also inhibited activity of the constitutively active IRF3-5D mutant [198,199]. 

The leader (L) protein of the cardioviruses of the *Picornaviridae* also interferes with IRF3 dimerisation. In fact, at first, the L protein of encephalomyocarditis virus (EMCV) was reported for its ability to generally disturb trafficking of proteins between the cytoplasm and nucleus [200]. After infection with a related strain, *Mengovirus*, however, IRF3 still localized to the nucleus. To inhibit type I IFN transcription nonetheless, the mengovirus L protein antagonizes dimerisation of IRF3 downstream of its phosphorylation [201]. Due to the observation of IRF3 translocation without dimerisation, the authors suggested that importin-mediated transport might be a requirement for dimerisation of IRF3, which would take place within the nucleus, which is in line with nuclear translocation of monomeric IRF3, as discussed above (see Section 2.3). The L protein from a different cardiovirus species, Theiler’s murine encephalomyelitis virus (TMEV), also interferes with the formation of IRF3 dimers, though it was first identified because it additionally inhibits the nuclear translocation of IRF3 [202,203]. This function depends on the Zinc finger motif of L [203], but whether L affects preceding phosphorylation of IRF3 as well remains undetermined. The strong impact of L on the antiviral innate immune response is additionally accounted for by the inhibition of mRNA export from the nucleus which is achieved by L-mediated phosphorylation of nucleoporin 98 [203].

In contrast to the yet unclear molecular mechanism of ML and L, an interesting detail of dimerisation antagonism was described for the serine/threonine kinase US3 of HSV-1 [204]. US3 phosphorylates IRF3 at S175, and this hyperphosphorylation interferes with both dimerisation and nuclear translocation of IRF3. As also activity of IRF3-5D could be inhibited by co-expressed US3 in a Luciferase-based reporter assay, the effect of US3 seems independent of the activation status. This adds a further phosphate group to the posttranslational modifications that modulate IRF3 activity and raises the question whether the cardiovirus L proteins might also interfere with the formation of IRF3 dimers via small modifications, just as TMEV L mediates inhibition of the nuclear pore protein by phosphorylation [203].

### 5.2. Selected Cast Only—Inhibition of Nuclear Translocation

The V protein of simian virus 5 (SV5) was the first reported antagonist of IRF3 translocation [205], followed by the multifunctional NS1 protein of Influenza B virus (IBV) [206] and the L protein of TMEV [202]. Again, these early studies did not yet dissect which form of IRF3 is targeted because the role of the essential modifications was not known in detail. Later reports of viral antagonists of IRF3 translocation then included an assessment of the activation state of IRF3, and some were found to specifically target IRF3 after its activation. For example, HSV-1 initially stimulates activation of IRF3 upon entry but subsequently inhibits IRF3 activity by means of newly synthesized infected cell polypeptide 0 (ICP0) [207]. During the viral replication cycle, a part of the multifunctional ICP0 molecules localises to the cytoplasm and inhibits the translocation of activated IRF3 into the nucleus in order to shorten the ongoing IRF3-dependent innate immune response. Curiously, while this cytosolic function of ICP0 is independent of the activity of its E3 ubiquitin ligase domain, the functional host proteasome is required for the localisation of ICP0 to the cytoplasm [207].

The inhibitory function on the innate immune response by the V protein of Sendai virus (SeV) was first accounted for by the interaction of V with the RNA sensor melanoma differentiation-associated protein 5 (MDA5) [208]. However, SeV infection is predominantly detected by RIG-I [209,210,211], and SeV infection still inhibits IFNβ production in MDA5-knockout mice [212]. For this reason, the group of Sakaguchi continued their studies and discovered that the SeV V protein interacts with IRF3 as well as IRF3-5D in the cytosol and inhibits nuclear translocation. In line with this, also the V proteins of the related measles virus (MeV) and Newcastle disease virus (NDV) of the *Paramyxoviridae* interact with IRF3 and interfere with its *trans*-activation activity, though MeV V seems to target IRF3 after nuclear translocation [212]. 

In contrast, the non-structural protein NS5 of the Flavivirus Japanese encephalitis virus (JEV) suppresses import of IRF3 indirectly by interacting with the nuclear transport proteins KPNA3 and KPNA4 [213]. This interaction competitively blocks the interaction with the native cargo of the importins, including the transcription factors IRF3 and p65. Similarly, the NS3/4A protease of the related hepatitis C virus (HCV) triggers cleavage of importin β1 (IPOB, also KPNB1) and in this way inhibits the transport of IRF3 into the nucleus to interrupt IFNβ production [214].

### 5.3. Gate-Crashers Barge in—Inhibition of the IRF3-CBP/p300 Holocomplex Formation

Once IRF3 has reached the nucleus, it interacts with the coactivator CBP/p300 to obtain the potential to induce transcription. When studies concerned with the molecular requirements for the induction of type I IFNs were still on their way, the critical role of the coactivators for IRF3 activity was underlined by the observation that the antagonistic activity of adenovirus (AdV) protein E1A on IRF3-mediated stimulation was dependent on the interaction of E1A with CBP [176]. The inhibition of *IFNB1* transcription by E1A could be competed with by overexpression of CBP/p300 but was not observed for a mutant of E1A that was defective in p300-binding. Mechanistically, these observations implied that E1A competed with IRF3 for binding to the essential coactivator [176]. 

To interfere with the productive association of IRF3 and CBP/p300, the viral antagonists NS1 of human respiratory syncytial virus (RSV) [215], E6 protein of human papillomavirus 16 (HPV-16) [175,216] and the tegument protein VP16 of HSV-1 target both proteins simultaneously [217]. In contrast, the kinase ORF36 of murine gammaherpesvirus 68 (MHV68) specifically interacts only with activated IRF3 in the nucleus to interfere with the association of coactivators [218]. This protein is highly conserved, implying similar functions of its homologue in Kaposi’s sarcoma-associated herpesvirus (KSHV), though the conserved kinase activity is not required for this function [218]. 

A striking example for the exploitation of structural homology by viruses are the viral homologues of IRFs, termed vIRFs, which are encoded by some herpesviruses to interfere with the activity of host IRFs. For details on the interplay of vIRFs and host IRFs and their implications for the viral replication style, please refer to the recent review by Myoung and colleagues [219]. Shortly, vIRF1 of KSHV binds to IRF3 as well as p300 and in this way obstructs formation of the active holocomplex of IRF3 and CBP/p300 [220,221,222]. Additionally, KSHV deploys vIRF2 to target IRF3-driven gene expression [223]. Similar to vIRF1, also the virion-associated vIRF R6 of the gammaherpesvirus rhesus macaque rhadinovirus (RRV) inhibits *IFNB1* expression by targeting CBP and competing with phosphorylated IRF3 for binding [224]. As tegument proteins, RRV R6 and also HSV-1 VP16 are released into the host cell alongside the entering virus and can directly take action to interfere with IRF3 activity before a potent type I IFN response can be mounted.

Another strategy to hinder the participation of the essential coactivator CBP is applied by several RNA viruses. The non-structural protein 1 (nsp1) subunits nsp1α of murine lactate dehydrogenase-elevating virus (LDV) [225], nsp1γ of simian haemorrhagic fever virus (SHFV) [225] and nsp1α of porcine reproductive and respiratory syndrome virus (PRRSV) [225,226,227] of the *Arteriviridae* family as well as nsp1 of porcine epidemic diarrhoea virus (PEDV) [228] of the *Coronaviridae* all suppress *IFNB1* expression by specifically targeting CBP in the nucleus and mediating its proteasome-dependent degradation. This approach implies that these viruses do not require the activity of CBP to express their own genome, which is in line with their cytoplasmic replication.

### 5.4. At the Eleventh Hour—Inhibition of DNA Binding

The final aim of IRF3 activation is to enable its binding to recognition motifs within the IFNβ enhancer in order to allow assembly of the IFNβ enhanceosome and induce transcription. Again, viral antagonists apply different mechanisms to interfere at this level. The NP1 protein of human bocaparvovirus (BoV) and US1 of HSV-2 prevent DNA binding by interaction with the DBD of IRF3 themselves [229,230]. Further, NSs of Sandfly fever Sicilian virus (SFSV) of the *Bunyaviridae* family was recently reported to apply this strategy to prevent association of IRF3 with the IFNβ promoter [231]. Wuerth and colleagues demonstrated that NSs specifically interacts with IRF3 via the DBD and by this masking competes for DNA binding in a dose-dependent manner. 

Several other strategies were identified in the *Herpesviridae* family: (i) The nuclear share of the HSV-1 protein ICP0 relocalises IRF3 and CBP/p300 in the nucleus to special nuclear structures, thereby sequestering them away from their site of activity [232]. Additionally, this mediates deactivation and promotes degradation of IRF3. The ICP0 variant of bovine herpesvirus 1 (BHV-1) also interacts with p300, but in contrast to HSV-1, this interaction hijacks the acetyltransferase activity of the coactivator and activates expression from viral promoters [233]. (ii) The kinase BGLF4 of Epstein–Bar virus (EBV) directly interacts with IRF3 and phosphorylates several residues between DBD and IAD [234]. Without affecting dimerisation or association with CBP, this modification prevents binding of IRF3 to DNA. (iii) KSHV latency-associated nuclear antigen 1 (LANA-1) competes with IRF3 for binding to the PRDIII-I region and in this way interferes with stable association of IRF3 to the target DNA [235]. The DNA polymerase subunit UL44 of HCMV was recently reported to act in a similar way [236]: Upon HCMV infection, exogenously expressed UL44 could be demonstrated to associate with the IFNβ promoter region in a chromatin immunoprecipitation assay and in parallel UL44 interfered with binding of the central transcription factors IRF3 and p65. Additionally, UL44 interacts with both transcription factors irrespective of their phosphorylation status. (iv) The KSHV protein K-bZIP binds to the IFNβ promoter region itself and prevents binding of IRF3 [237], but differently from LANA-1 and UL44, K-bZIP weakly induces gene expression while it prevents a high and detrimental activation of the type I IFN response. 

In contrast, the NSs protein of Rift Valley fever virus (RVFV) applies a more indirect approach to inhibit binding of the IRF3-CBP/p300 holocomplex [238]. NSs interacts with the host factor Sin3A-associated protein 30 (SAP30), which belongs to the Sin3A/NCoR/HDACs repressor complexes. In turn, SAP30 interacts with the transcription factor YY1, and this interaction enables recruitment of NSs and SAP30 to the murine IFNβ promoter region. Finally, the presence of the NSs-SAP30-YY1 complex inhibits recruitment of CBP and thus transcriptional activation of *IFNB1*.

## 6. To Be Continued—Perspective

Since the beginning of the detailed characterisation of IRF3 activation more than 20 years ago, we have gained profound insight into the mechanisms enabling the *trans*-activation activity of a transcription factor with low intrinsic binding affinity for its specific recognition motif and into the intricate network of interactions that allow its participation in the induction of *IFNB1* expression. Still, the fundamental principles were mostly characterised in vitro and sometimes biased from prior assumptions, rendering the sequence of events as it occurs in our cells incomplete. To reveal the full dynamics of the events summarised above, experiments with living cells will be required. The many studies reporting factors that affect IRF3 dimerisation and/or translocation and/or *trans*-activation activity highlight the importance of pinpointing the modulatory mechanism as precisely as possible in future work. At the same time, the diversity of methods applied in the studies summarised here demonstrates that nowadays, dissection of the exact level of intervention with IRF3 activity is within reach. 

As delineated in this review, the regulation of IRF3 in the context of *IFNB1* expression in itself presents us with a multi-layered circuit of promoting and dampening modulations. The ever-expanding catalogue of host and viral modulators of IRF3 activity not only reflects the key role of this signalling pathway in the mammalian arsenal of antiviral defence mechanisms, but it further alludes to the possibility to use specific mechanisms for medical intervention. Directed manipulation of host modulators could present a novel approach to stimulate the host organism to overcome an infection by its own resources or to reduce excessive IFN activity to healthy levels. More information on how to achieve this will surely emerge from the unceasing identification and characterisation of novel IRF3 modulators deployed by host and virus.

Nevertheless, the multitude of additional genes targeted by IRF3 signifies that we are still just beginning to understand the true complexity of IRF3 fine regulation. In addition to the ongoing characterisation of the direct involvement in the expression of ISGs, powerful in silico and sequencing approaches of recent years revealed a growing list of novel targets, including virus-inducible RNAs [107,239]. Moreover, some of the identified host modulators hint at regulation of IRF3 activity dependent on signalling of other pathways to enable the incorporation of further parameters into the type I IFN response [39,148,240]. To successfully harness this powerful system, an in-depth comprehension of all contributing factors will be necessary. 

## Figures and Tables

**Figure 1 viruses-12-00733-f001:**
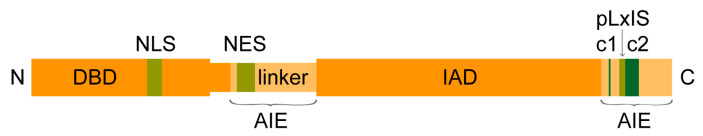
Schematic representation of the transcription factor interferon regulatory factor 3 (IRF3)**.** IRF3 consists of an N-terminal DNA-binding domain (DBD) linked by a flexible region to the C-terminal IRF association domain (IAD). In the latent state, a part of the linker and the most C-terminal portion form auto-inhibitory elements (AIEs). Phosphorylation of two serine-rich clusters (c1 and c2) induces conformational rearrangements of the AIEs and frees the IAD to participate in protein–protein interactions with coactivators and for dimerisation via the phosphorylated pLxIS motif (p: hydrophilic, x: any amino acid). The protein further contains a nuclear localization signal (NLS) and a nuclear exit signal (NES) that enable constitutive shuttling between the cytosol and nucleus.

**Figure 2 viruses-12-00733-f002:**
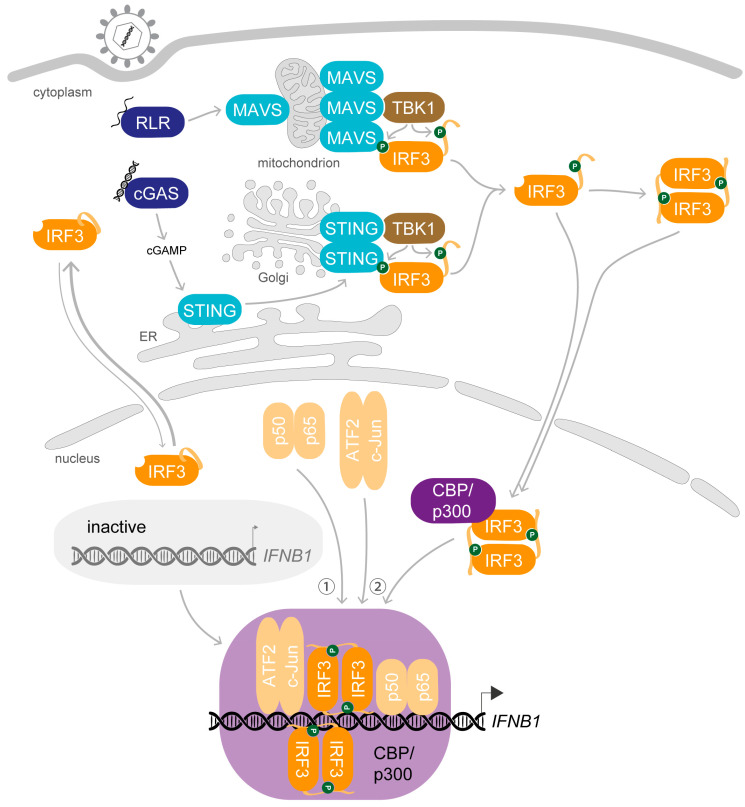
IRF3 activation induces transcription of *IFNB1* upon viral infection. In resting cells, latent IRF3 shuttles between the cytosol and nucleus. Upon viral infection, viral nucleic acids are recognized by RNA or DNA sensors in the cytosol. Activation of retinoic acid-inducible gene I (RIG-I)-like receptors (RLRs) like RIG-I induces aggregation of the mitochondrial adaptor protein mitochondrial antiviral signalling protein (MAVS). Detection of DNA by the DNA sensor cyclic GMP-AMP (cGAMP) synthase (cGAS) activates production of the second messenger cGAMP, which in turn induces dimerisation of the adaptor protein stimulator of interferon genes (STING) and its translocation from the endoplasmic reticulum (ER) to the Golgi apparatus. Higher-order structures of the adaptor molecules recruit the kinase TANK-binding kinase 1 (TBK1) which leads to their TBK1-mediated phosphorylation. IRF3 is recruited to this platform and gets phosphorylated by TBK1 at key residues in the AIE, relieving the auto-inhibition. Activated IRF3 heterodimerises and associates with the coactivators CBP/p300 after translocation into the nucleus, yielding a holocomplex with *trans*-activation potential. In parallel, the heterodimeric transcription factors p50-p65 (NF-κB) and ATF2-c-Jun (AP-1) are activated and enter the nucleus. First, p50-p65 is recruited to the enhancer element upstream of the *IFNB1* gene, followed by ATF2-c-Jun and the IRF3-CBP/p300 holocomplex. The assembled IFNβ enhanceosome promotes recruitment of the basal transcription machinery for the expression of *IFNB1*.

**Figure 3 viruses-12-00733-f003:**
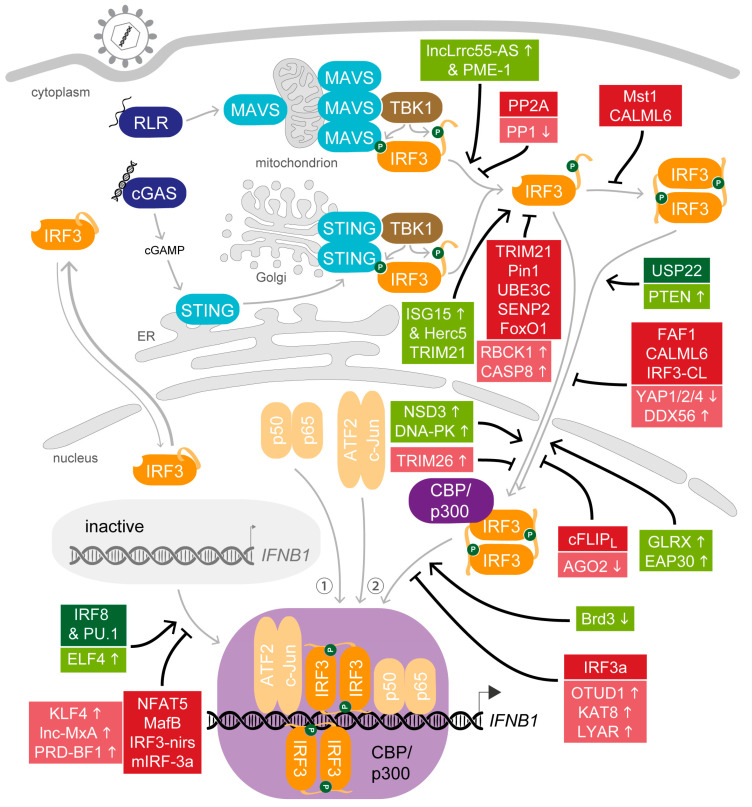
Host modulators of IRF3 activity. Green and red boxes indicate constitutively positive and negative modulators of steps leading to IRF3-mediated *IFNB1* expression, respectively. Small arrows mark up- (↑) or down-regulation (↓) of the supporting (faded green boxes) or inhibitory (faded red boxes) role of host factors upon activation of antiviral signalling.

**Figure 4 viruses-12-00733-f004:**
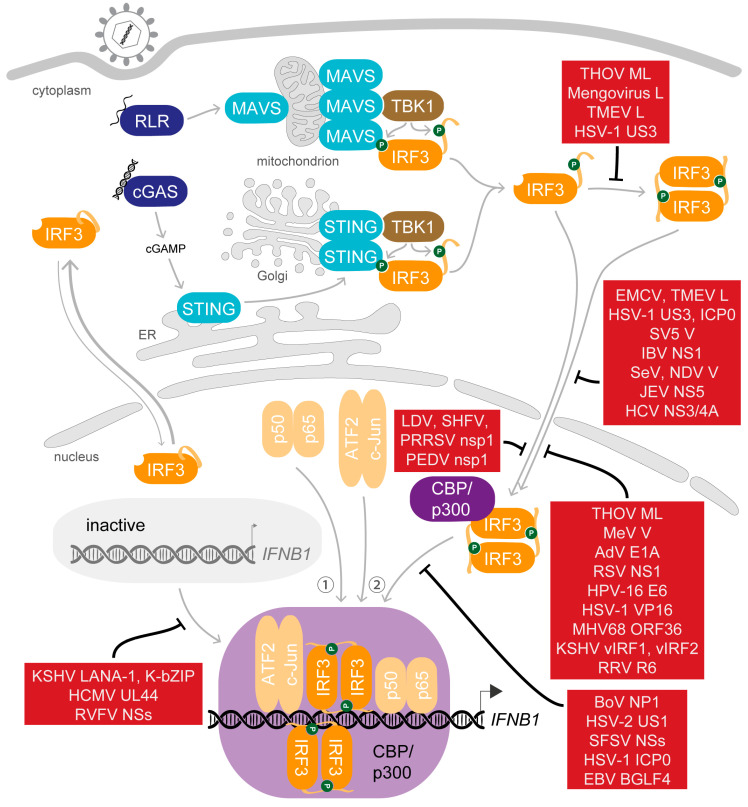
Viral antagonists of phosphorylated IRF3. Viral inhibitors of IRF3-dependent *IFNB1* expression are listed in red boxes, with blind-ended lines indicating the step of intervention. Not shown are viral factors targeting the initial phosphorylation step and factors mediating IRF3 degradation.

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
