# Peer review of "Of Keeping and Tipping the Balance: Host Regulation and Viral Modulation of IRF3-Dependent IFNB1 Expression"

_viruses, 2020, doi:10.3390/v12070733_

Round 1

Reviewer 1 Report

Congratulations to the authors. An excellent manuscript and a herculean effort to pull it altogether. I only have a couple of questions/comments that could be discussed in the manuscript.

Given that NF-KB is also a regular target of viral inhibitors what are the consequences of this on enhanceosome formation. The authors discuss the importance of the p50-p65 subunit in IFNB-1 transcription but not the net result of its inhibition on IRF3 dimers being loaded onto the enhancer to initiate transcription.

Second, what does viral inhibition look like from a temporal perspective? How quickly can viruses inhibit this process (can they block initial rounds of enhanceosome formation) or are they more focused on inhibiting the later stages where IRF7 players a larger role than NFKB? And how does this relate to pathogenesis – while viruses all seem to be great modulators of innate immune activation almost all (especially the RNA viruses like flu, RSV, picornas) are cleared from immunocompetent individuals. How can this be the case when they encode such potent inhibitors – is it simply a case of them delaying the IRF3 response as long as possible rather than totally blocking it?

Minor points:

Abstract: For internet search engine reasons only, it might be worth naming the five transcription factors (line 15)

Throughout the manuscript I wasn’t clear what “(see 0)” was referring to? Please clarify. Also, throughout I would include the article “the” before “cytosol/cytoplasm and nucleus” when using this construct.

Line 62: I would prefer “viruses are engaged”, rather than they engage

Line 114: is IRF7

Line 134: prime cellular

Line 139: phosphorylated

Line 142: knowledge of the

Line 143: IRF3 has been greatly

Line 215: assess

Line 231: experiments from several groups

Line 234: that the presence

Line 251: Due to these discrepancies

Line 254: as a transcription factor

Line 386: transcription form the remaining

Line 413: most studies have thus far focused on the murine system.

Line 426: required?

Line 435: nascent instead of “commencing”?

Line 510: Contrasting this,

Line 555: Irrespective

Line 557: Regulation of auto-inhibition aside

Line 558: also at subsequent steps

In addition, a few sentences might need double checking for clarity and/or succinctness.

Line 100, “For simplicity…”

Line 194, “Since its…”

Line 209, “These large…”

Line 218, Considering that…”

Line 249, “Several group…”

Line 281, “From just shortly…”

Line 446, “Especially mechanims…”

Figure 3 legend

Line 531: “Especially macrophages…”

Author Response

Response to Reviewer 1 Comments

Many thanks to the reviewer for going through our manuscript so thoroughly. The reviewer posed very interesting questions, part of which were discussed internally while reviewing the literature and chose to exclude due to lack of studies. Further, we thank the reviewer for giving valuable feedback to improve the comprehensibility and the use of the language.

Questions/comments by the reviewer:

Congratulations to the authors. An excellent manuscript and a herculean effort to pull it altogether. I only have a couple of questions/comments that could be discussed in the manuscript.

Q1: Given that NF-KB is also a regular target of viral inhibitors what are the consequences of this on enhanceosome formation. The authors discuss the importance of the p50-p65 subunit in IFNB-1 transcription but not the net result of its inhibition on IRF3 dimers being loaded onto the enhancer to initiate transcription.

Response Q1: Thank you for this comment. This is a very interesting question but opens the even wider field of NF-kB modulation which we chose to not go into with this review. As you mentioned, NF-kB is also a prominent target of viruses. While viral antagonists oftentimes inhibit functions of NF-kB, NF-kB is also exploited due to its pro-survival functions to ensure survival of the host cell after infection. Subsequent to the dissection of the important role of p50-p65 in the formation of the enhanceosome in the very early steps after infection by the group of Beg (reviewed in 2011), we did not find any follow-up studies looking into the temporal succession of interaction events leading to IFNB1 expression. Given its importance in mediating interaction of IRF3 to DNA together with CBP/p300, we would speculate that inhibition of p50-p65 activity, or more precisely inhibition of DNA binding by the dimer, also inhibits the recruitment of IRF3 dimers to the enhanceosome (as long as the virus does not provide a very strong stimulus, as highly active IRF3 can drive IFNB1 expression by itself). However, we did not find reports following up on the effect of NF-kB inhibition on the ensuing formation of the enhanceosome. Possibly, this supposed domino effect of inhibition of NF-kB binding translating to inhibition of IRF3 binding might account for reports of antagonists inhibiting the activity of both transcription factors. We are curious about future studies tracking the interactions of the individual transcription factors and the further machinery within the cell. Including viral inhibitors of specific transcription factors to such experimental setups certainly is an interesting tool to precisely define the sequence and temporal requirement of events.

Q2: Second, what does viral inhibition look like from a temporal perspective? How quickly can viruses inhibit this process (can they block initial rounds of enhanceosome formation) or are they more focused on inhibiting the later stages where IRF7 players a larger role than NFKB? And how does this relate to pathogenesis – while viruses all seem to be great modulators of innate immune activation almost all (especially the RNA viruses like flu, RSV, picornas) are cleared from immunocompetent individuals. How can this be the case when they encode such potent inhibitors – is it simply a case of them delaying the IRF3 response as long as possible rather than totally blocking it?

Response Q2: Also an excellent and even more complex question with many remaining question marks. Going through the reports of viral antagonists, we tried to include cues about the temporal aspect of the inhibitory function if this was mentioned in the respective publication. For example, some herpesviral inhibitors are delivered with the incoming particle as part of the tegument and can immediately counter IRF3 activity and formation of the enhanceosome. Others, like ICP0, were explicitly described to be important for inhibition of IRF3 during ongoing infection, meaning that inhibition of enhanceosome formation with IRF3 is still important during ongoing infection and not solely shifted to IRF7.

Herpesviruses are a good example to demonstrate that the interplay of many factors defines the final response. Although herpesviruses enter the cell with numerous tegument proteins which counteract PRR-mediated signalling, and on top encode a vast array of antagonists further targeting steps of immune signalling pathways, activation of a strong innate immune response upon viral infection is still evident. Moreover, often deletion of just one viral modulator from the herpesviral genome can lead to a strong attenuation in vivo, which opens the question why the virus encodes so many of them. A possible explanation for this could be cell type-dependent effects of viral antagonists and the requirement for a finely balanced interplay with host factors during the course of infection. Antagonists expressed by RNA viruses are often non-structural proteins that have to be translated first, however, the virus still inhibits the immune responses to an extent that establishment of infection and spread to the next host is often successful before it is cleared by the full blown host immune response. Together, these examples could indicate that the temporal factor of viral inhibition aligns with the life style of the virus: For herpesviruses which want to achieve a chronic infection, the immediate as well as ongoing interference with IFN production is essential to establish latency. For RNA viruses passing through, it may be enough to dampen IRF3 activity until enough progeny could be produced to reach the next host – there might just be no necessity for RNA viruses to constantly abolish the IFN response.

Minor Points

Point 1: Abstract: For internet search engine reasons only, it might be worth naming the five transcription factors (line 15)

 Response 1: We decided not to name the other factors in the abstract because only IRF3 is discussed in detail in the review. Internet search engines can probably find far more relevant literature on AP-1 and NF-kB and their modulation.

Point 2: Throughout the manuscript I wasn’t clear what “(see 0)” was referring to? Please clarify. Also, throughout I would include the article “the” before “cytosol/cytoplasm and nucleus” when using this construct.

Response 2: Using the search option of word we cannot find “(see 0)” in the manuscript now – probably these are the cross-references we put in some paragraphs so a reader interested in a specific topic can easier find the relevant section. (For example, in line 98 “(see 2.1)” refers to the section of IRF3’s protein-protein interactions.) When formatting is changed, the references can be lost or get erroneous – hopefully, this will work for the final (print) version!

The article was added before “cytosol/cytoplasm and nucleus” as suggested.

Point 3:

Line 62: I would prefer “viruses are engaged”, rather than they engage

Line 114: is IRF7

Line 134: prime cellular

Line 139: phosphorylated

Line 142: knowledge of the

Line 143: IRF3 has been greatly

Line 215: assess

Line 231: experiments from several groups

Line 234: that the presence

Line 251: Due to these discrepancies

Line 254: as a transcription factor

Line 386: transcription form the remaining

Line 413: most studies have thus far focused on the murine system.

Line 426: required?

Line 435: nascent instead of “commencing”?

Line 510: Contrasting this,

Line 555: Irrespective

Line 557: Regulation of auto-inhibition aside

Line 558: also at subsequent steps

In addition, a few sentences might need double checking for clarity and/or succinctness.

Line 100, “For simplicity…”

Line 194, “Since its…”

Line 209, “These large…”

Line 218, Considering that…”

Line 249, “Several group…”

Line 281, “From just shortly…”

Line 446, “Especially mechanims…”

Figure 3 legend

Line 531: “Especially macrophages…”

Response 3: These points were addressed accordingly in the manuscript. The verb “(to) assess” in line 215 is used in past tense because it refers to past/closed studies. Regarding the note for line 426 (“required?”) and assuming that this is not a phrasing suggestion: We found these points interesting but tried to phrase them carefully as the mechanism underlying the effect of the YY proteins was not formally demonstrated so far for both the murine and the human system.

Reviewer 2 Report

In this review article, the authors describe in detail the regulatory mechanism of IRF3-mediated activation of type I IFN genes, focusing on positive- and negative-regulation by host factors and inhibitory effects by viral proteins. The manuscript is well-written and is useful for a better understanding of antiviral innate immune responses.

Minor Points

1) In the figures, the correct spelling of "michondrium" could be "mitochondrion" in English.

2) It would be appropriate to unify the description of lysine residues with “Lys” or “K”.

Author Response

Response to Reviewer 2 Comments

We highly appreciate that the reviewer read the manuscript very carefully and pointed out inconsistencies in the details which had escaped our notice.

In this review article, the authors describe in detail the regulatory mechanism of IRF3-mediated activation of type I IFN genes, focusing on positive- and negative-regulation by host factors and inhibitory effects by viral proteins. The manuscript is well-written and is useful for a better understanding of antiviral innate immune responses.

Minor Points

Point 1: In the figures, the correct spelling of "michondrium" could be "mitochondrion" in English.

 Response 1: The spelling was corrected in all figures.

Point 2: It would be appropriate to unify the description of lysine residues with “Lys” or “K”.

Response 2: Amino acid residues are now consistently labelled using the single letter code.